# A Data-Driven Framework for Digital Transformation in Smart Cities: Integrating AI, Dashboards, and IoT Readiness

**DOI:** 10.3390/s25165179

**Published:** 2025-08-20

**Authors:** Ángel Lloret, Jesús Peral, Antonio Ferrández, María Auladell, Rafael Muñoz

**Affiliations:** 1Language and Information Systems Group, Department of Software and Computing Systems, University of Alicante, 03690 Alicante, Spain; lloret@ua.es (Á.L.); antonio@dlsi.ua.es (A.F.); maa104@gcloud.ua.es (M.A.); rafael@dlsi.ua.es (R.M.); 2Lucentia Research Group, Department of Software and Computing Systems, University of Alicante, 03690 Alicante, Spain

**Keywords:** digital transformation, public administration, artificial intelligence, smart city, IoT

## Abstract

Digital transformation (DT) has become a strategic priority for public administrations, particularly due to the need to deliver more efficient and citizen-centered services and respond to societal expectations, ESG (Environmental, Social, and Governance) criteria, and the United Nations Sustainable Development Goals (UN SDGs). In this context, the main objective of this study is to propose an innovative methodology to automatically evaluate the level of digital transformation (DT) in public sector organizations. The proposed approach combines traditional assessment methods with Artificial Intelligence (AI) techniques. The methodology follows a dual approach: on the one hand, surveys are conducted using specialized staff from various public entities; on the other, AI-based models (including neural networks and transformer architectures) are used to estimate the DT level of the organizations automatically. Our approach has been applied to a real-world case study involving local public administrations in the Valencian Community (Spain) and shown effective performance in assessing DT. While the proposed methodology has been validated in a specific local context, its modular structure and dual-source data foundation support its international scalability, acknowledging that administrative, regulatory, and DT maturity factors may condition its broader applicability. The experiments carried out in this work include (i) the creation of a domain-specific corpus derived from the surveys and websites of several organizations, used to train the proposed models; (ii) the use and comparison of diverse AI methods; and (iii) the validation of our approach using real data. Based on the deficiencies identified, the study concludes that the integration of technologies such as the Internet of Things (IoT), sensor networks, and AI-based analytics can significantly support resilient, agile urban environments and the transition towards more effective and sustainable Smart City models.

## 1. Introduction

The penetration of digital technologies is a major challenge in today’s economic, political, social, and scientific domains. Its impacts have profoundly influenced society, leading to changes in everyday life—such as the digitization of public services, the widespread use of mobile applications, and new forms of communication and work—thus advancing the process known as digital transformation (DT). DT plays a key role in bridging the structural, strategic, and technological changes required to meet the demands of the contemporary digital era [1]. This process involves aligning traditional working practices and corporate or administrative systems with emerging technologies, fostering innovation aimed at transforming organizational products and processes, and addressing both potential and challenges [2].

In the current context of accelerated urbanization and climate challenges, Smart Cities have emerged as a strategic response to the DT of municipalities. While various local initiatives have driven DT at the municipal level, it is essential to frame these efforts within a broader comparative perspective that enables the identification of common patterns, challenges, and opportunities at the international level. In this regard, recent studies have shown that the maturity of Smart City platforms—assessed through structured models such as the ISO 37123 standard—is a key factor in advancing toward resilient, sustainable, and intelligent urban environments [3]. Unlike approaches focused exclusively on national or regional policies, this work introduces an innovative methodology that combines structured and unstructured data with Artificial Intelligence (AI) techniques, enabling automated and scalable evaluation of DT in local public administrations. This methodological approach not only addresses the limitations of traditional models but also contributes to the international literature by offering a replicable and adaptable framework for diverse institutional contexts.

Over the past decades, DT in the public sector has become a strategic priority for the European Union, which seeks to modernize the administration of its Member States through common policies [4,5,6]. However, the lack of alignment between digital public administration systems and private sector technologies can hinder effective communication and service delivery. In some cases, digital workflows are not fully integrated, leading to continued reliance on paper-based processes, which undermines public trust in innovation and weakens the credibility of public administrations as role models aligned with EU digital standards. Moreover, limited interoperability and the absence of real-time data sharing tools can prevent the implementation of smart urban management solutions, reducing the societal acceptance of emerging technologies and slowing down progress toward ESG (Environmental, Social, and Governance) goals and the UN SDGs (United Nations Sustainable Development Goals). To this end, the European Commission has promoted several initiatives to ensure this process is implemented correctly, in line with clear guidelines and objectives. The most notable of these are the eGovernment Action Plan [7] and the Digital Decade Policy Programme 2030 [8].

The eGovernment Action Plan aims to improve the quality, accessibility, and efficiency of digital services provided by European public institutions. It promotes the adoption of technologies such as AI, big data, and cloud computing, which help streamline administrative procedures and data management, ultimately improving the citizen experience.

Meanwhile, the Digital Decade Policy Programme 2030 provides tools to accelerate digitalization in the public sector across Europe. A key element is the Digital Innovation Lab (iLab), which offers resources to help public administrations adopt innovative technologies and efficiently manage their digital projects. Among its resources are the Data Innovation Repository [9] and the Data Innovation Toolkit [10]—two essential tools that not only support DT efforts in public institutions but also ensure data security and the continuous improvement of public service delivery.

Aligned with these European strategies, Spain has undergone significant DT, accelerating economic, social, and administrative development. This is reflected in the expansion of digital public services, the implementation of nationwide digital identity systems, and the increased use of AI-based tools in administrative processes. This process has been led by institutions such as the State Agency for Digital Administration (Agencia Estatal de Administración Digital, AEAD—https://administracionelectronica.gob.es/pae_Home/pae_Organizacion/AEAD.html (visited on 4 July 2025)), and the State Secretariat for Digitalization and Artificial Intelligence (Secretaría de Estado de Digitalización e Inteligencia Artificial, SEDIA—https://digital.gob.es/ministerio/organigrama_organos/SEDIA.html (visited on 4 July 2025)), which act as both designers and coordinators of the national digital strategy. Moreover, the COVID-19 pandemic significantly accelerated the adoption of digital technologies, granting them an even more prominent role in public sector management.

Within this broader national context, this study focuses on how DT is also advancing at the local level, where municipal governments play a crucial role in implementing DT policies aimed at promoting transparency, efficiency, and citizen engagement [11,12,13,14,15].

DT in local governments is a complex process that goes beyond the mere adoption of technological tools. It requires a reconfiguration of organizational models, administrative processes, and—most importantly—the internal culture of public institutions. DT should be understood as an opportunity to strengthen principles of transparency, efficiency, and civic participation, while reorienting public management towards a more open, innovative, and citizen-centered approach [16].

In light of these developments, DT has become a top priority in public administrations, as it enables the provision of high-quality services and enhances citizen satisfaction. However, in many cases, there is a noticeable lack of capacity to measure and assess progress in DT. This limitation hinders the identification of areas for improvement and the ability to make data-driven decisions aimed at enhancing service delivery.

A key component of the DT process is the Internet of Things (IoT), which, when combined with cloud computing, opens up new opportunities to enhance the quality of public services [17]. As a disruptive technology, IoT has the potential to transform local public administrations significantly. In particular, edge computing seeks to address the processing demands of IoT applications by bringing data sources closer to cloud services at the edge of the network, thereby reducing latency and improving overall performance [18]. This enables the creation of real-time information networks that can be leveraged to improve the efficiency, transparency, and quality of government services, contributing directly to the DT of municipalities into Smart Cities.

A Smart City can be conceptualized as a digitally enhanced urban environment that applies interconnected systems and sensor networks. These technologies enable the continuous acquisition, processing, and dissemination of data to support evidence-based decision-making and service innovation. This paradigm is grounded in the integration of advanced Information and Communication Technologies (ICT), including IoT infrastructures, cloud and edge computing, and secure data management frameworks. These technologies enhance the efficiency, sustainability, and responsiveness of urban services. Smart City initiatives typically span strategic and data-sensitive domains such as energy, mobility, environmental monitoring, healthcare, governance, and public safety, with the overarching goal of improving the quality of life for all stakeholders through data-driven urban intelligence [19].

Two essential features characterize Smart Cities [20,21]. First, a distributed collection of sensors continuously monitors urban activity by capturing and measuring diverse indicators from any location, at any time. Second, data analysis tools and applications extract meaningful insights from this information, enabling informed decision-making, proactive problem-solving, and efficient resource coordination. Intelligent sensing devices can generate large volumes of data, and big data capabilities enable the creation of data-driven Smart Cities and the implementation of sustainable urban initiatives. Thus, to fully realize the potential of the Smart City model, it is essential to ensure the effective deployment of technical infrastructure (including interoperable systems and intelligent communication networks), robust data security mechanisms, and the protection of sensitive personal data—especially given the increasing risks of cyberattacks. Furthermore, user education and stakeholder involvement are critical to building public trust, fostering acceptance, and ensuring sustainable and inclusive digital transformation.

After analyzing the key characteristics and implications of DT—particularly within local public administrations—and reviewing existing Smart City initiatives [22], the study concludes that, while the technologies required for DT already exist, there is, according to the result of the study’s literature review, no established method for automatically assessing the DT level of an organization. Therefore, the main objective of this paper is to propose an innovative methodology for evaluating DT in public sector organizations by combining traditional assessment approaches with AI techniques. The proposed methodology is validated through a real-world case study. The benefit of this approach lies in its ability to analyze the actual digital maturity of an organization automatically and to suggest data-driven actions for improvement, ultimately enhancing the quality of public services delivered to citizens.

The summarized main contributions are the following:The design of a taxonomy for Smart Cities that identifies the key components of DT, framed within a comprehensive 360-degree model.The proposal of a holistic methodology to automatically assess the DT status of local public administrations, integrating multiple organizational dimensions.The creation of a domain-specific corpus compiled from the official websites of various local public administrations, as well as from surveys conducted among relevant stakeholders, with the aim of serving as a foundation for the analysis and training of AI models. Furthermore, the corpus has been made openly available to the scientific community to facilitate further research and experimentation.A comparative evaluation of traditional methods (e.g., surveys) and various AI models, conducted and validated through a real-world case study involving local public administrations.The design of interactive dashboards that support the extraction of insights and action plans to guide and enhance DT processes within public administrations.The development of a flexible methodology that can be adapted to other organizational domains, such as healthcare providers, educational institutions, or industrial companies.An analysis of the Smart City component—specifically city sensorization (e.g., for air quality, noise, and solar radiation)—highlighting its current underutilization and proposing strategic actions to reinforce its development.

The remainder of this paper is structured as follows. Section 2 reviews the most relevant previous work related to the assessment of DT. Section 3 presents the architecture proposed in this study. Section 4 describes a real-world case scenario in which the methodology is applied and evaluated through experimentation. Finally, Section 5 discusses the main findings, the practical implications of the model, and the limitations and future directions for research in this area.

## 2. State of the Art

The literature on DT assessment includes various indices and maturity models that evaluate how organizations adopt digital technologies across different dimensions. These approaches can be grouped into two categories: (1) indices specifically aimed at measuring DT as a holistic organizational process, and (2) models focused on assessing digital maturity through multidimensional evaluations.

Concerning the first group, the Digital Transformation Index (https://metarius.com/en/what-is-the-digital-transformation-index/ (visited on 4 July 2025)) is one of the most referenced models identifying core components such as technological infrastructure, business process digitalization, leadership and culture, privacy, digital customer experience, and data security. It provides organizations with a structured method for evaluating their digital progress and identifying areas for improvement, particularly in strategic planning.

In the corporate sector, ref.  [23] proposed an empirically grounded index composed of six weighted dimensions: strategic leadership, technological drive, organizational empowerment, environmental support, digital outcomes, and applied digital technologies. The model generates a composite score that facilitates comparative assessments of digital maturity across firms. Its design emphasizes scalability and support for data-driven decision-making in DT initiatives.

A different approach is offered by De la Peña and Cabezas [24], who introduced a conceptual formula to express DT as a combination of interdependent drivers. These include technology, customers, human factors, speed, value, organizational needs, and communications. Although the formula has not been empirically tested, it serves as a theoretical model to illustrate the complexity and multidimensionality of DT.

At the policy level, the European Commission developed the Digital Economy and Society Index (DESI) to monitor the digital progress of EU Member States. It assesses performance in key areas of the digital economy and society, helping to identify strengths and weaknesses and to guide public policy toward DT (https://digital-strategy.ec.europa.eu/en/policies/desi (visited on 4 July 2025)).

Until 2022, the DESI was structured around four key factors to evaluate its degree of success (https://digital-decade-desi.digital-strategy.ec.europa.eu/datasets/desi-2022/indicators (visited on 4 July 2025)): human capital, connectivity, integration of digital technologies in business, and the digitalization of public services. Each of these areas contributed equally to the overall index score. The DESI is based on data from Eurostat, national surveys, and specialized studies, and it has been widely used as a benchmark for evaluating digital advancement in the public sector. Although originally designed for individual organizations, the DESI has also been cited in studies focusing on local government digital strategies.

In addition to DT indices, the literature also offers a wide range of digital maturity models that assess overall corporate digital capabilities. These models typically evaluate multiple organizational dimensions—including leadership commitment to digital innovation, organizational culture, technological infrastructure, business processes, and customer orientation. Rather than producing a single score, maturity models aim to provide a diagnostic view of an organization’s strengths and gaps, guiding its path toward DT [25].

The InAsPro (Integrierte Arbeitssystemgestaltung in digitalisierten Produktionsunternehmen) model [26] was developed to evaluate digital maturity across different phases of the product lifecycle in manufacturing companies, such as development, production, assembly, and aftersales. It considers four inter-related dimensions—technology, organization, social aspects, and strategy. These dimensions refer, respectively, to the use and integration of digital tools, the structuring of work processes and responsibilities, the impact of digitalization on employee roles and collaboration, and the alignment of digital initiatives with long-term business objectives. The model applies a four-level maturity scale ranging from explorer to expert and enables organizations to identify gaps and prioritize digital initiatives through visual tools such as radar charts and maturity profiles.

The Digital Transformation Self-Assessment Maturity Model (DX-MM) model [27] offers a structured, self-assessment tool for organizations to evaluate their internal digital readiness. It focuses on four key dimensions—strategic alignment, organizational culture, digital competencies, and business process maturity—enabling a comprehensive self-assessment without relying on external audits. This model is especially useful in contexts where DT initiatives must be closely aligned with organizational strategy and operational capabilities. One of the model’s main contributions is its two-phase evaluation structure. The first phase assesses the maturity of business processes, while the second evaluates the alignment of those processes with the organization’s DT objectives. This approach ensures that both operational depth and strategic direction are taken into account. By integrating concepts from business process management and digital strategy, the DX-MM supports internal reflection, identifies digital capability gaps, and helps organizations prioritize their transformation efforts effectively.

Szelągowski and Berniak-Woźny [28] proposed an innovative framework linking DT assessment to business process maturity under specific organizational and strategic conditions. Traditional Business Process Management Maturity Models (BPMMMs) often fall short of capturing the complexity and knowledge intensity of modern business environments. To address this gap, the authors introduce a two-phase model. The first phase identifies critical success factors aligned with strategic objectives and classifies processes based on structure and knowledge intensity. The second phase applies tailored maturity assessments to each process group, offering a nuanced view of digital readiness. This approach integrates seamlessly into the BPM lifecycle and leverages tools such as process mining and AI to support both initial implementation and continuous improvement. It emphasizes that maturity assessments should not be isolated evaluations but should be embedded within broader strategic management practices. By adapting to process-specific characteristics and organizational context, the model captures not only levels of automation but also adaptability and learning capacity, making it a robust and context-aware tool for guiding DT.

Finally, Michelotto and Joia [29] introduce a model for assessing Organizational Digital Transformation Readiness (ODTR) based on a multidimensional maturity framework. This model incorporates technological, operational, leadership, human, and cultural dimensions, each evaluated through a six-level maturity scale adapted from Capability Maturity Model Integration (CMMI). It recognizes the dynamic interplay between internal capabilities and external pressures, framing DT as an ongoing and systemic organizational change process. It moves beyond technology-centric perspectives by incorporating human and cultural factors, thereby reflecting the systemic nature of organizational change in the digital era [30]. As such, it serves as a valuable tool for researchers and practitioners seeking to understand, measure, and enhance organizational readiness in increasingly digitalized environments.

### 2.1. Alignment with Sustainability Frameworks

The recent literature has highlighted the importance of aligning DT initiatives with broader sustainability and governance frameworks, such as the Environmental, Social, and Governance (ESG) criteria and the United Nations Sustainable Development Goals (SDGs) [31]. For instance, Gu et al. proposed a model for government ESG reporting driven by digital strategies in urban ecosystems, highlighting how ESG-aligned policies can enhance transparency and accountability in public service delivery [32]. Likewise, Smart City logistics and urban governance approaches are increasingly being assessed through an ESG lens to ensure that digitalization also supports environmental stewardship and social equity [33]. Smart City strategies are increasingly evaluated not only in terms of their technological advancements, but also by their ability to foster inclusive, transparent, and sustainable urban development [34,35].

In this context, digital tools, IoT infrastructures, and AI-based services are recognized as enablers for achieving multiple SDGs, including SDG 9 (Industry, Innovation and Infrastructure) and SDG 11 (Sustainable Cities and Communities). Parra-Domínguez et al. conducted a comprehensive literature review showing that Smart Cities contribute significantly to the mentioned SDGs [36]. Similarly, ESG criteria have become essential in assessing the societal and environmental impact of digital policies in both public and private sectors.

Collectively, these works support the view that our proposed methodology—which integrates AI techniques, real-time data analysis, sensor readiness, and dashboards—can serve as a tool for assessing DT in ways that are not only efficient but also aligned with global sustainability goals.

### 2.2. Findings and Contributions of Our Proposal

The analysis of the most relevant contributions (summarized in Table 1) reveals a clear gap in the availability of evaluation frameworks specifically tailored to the public sector. While many existing digital maturity models offer valuable insights into internal organizational readiness, they are predominantly designed for private or industrial contexts. As a result, their applicability to local public administrations is limited—particularly in terms of capturing the complexity and specificity of citizen-oriented services. Our approach incorporates dimensions that are specific to the municipal domain—such as Smart City platforms and smart tourism services—which are typically absent from generic models but are essential in local contexts. Furthermore, these models often lack integration with AI techniques and external data sources, which restricts their adaptability and limits their potential for continuous improvement.

In response to these limitations, this study proposes a novel weighted formula for assessing DT in local governments. The model represents a significant methodological advancement by combining structured data from municipal surveys with unstructured content extracted from official websites. This dual-source approach enhances the model’s ability to reflect both internal organizational dynamics and the external digital footprint of public institutions. This contribution not only fills a gap in the evaluation of public sector DT but also introduces a replicable and data-driven methodology that can support strategic decision-making and foster the development of more resilient, citizen-focused Smart Cities.

One area of concern consists of the selection and weighting of the index dimensions to compute the final score (a 70/30 distribution between general and context-specific components, which is grounded in the research team’s expertise and supported by informal expert consultations). The two-layer aggregation approach, while methodologically sound, may reduce the interpretability of individual dimension scores. This could hinder the identification of specific areas requiring strategic intervention. Unlike other models that provide diagnostic profiles or maturity levels, the proposed index yields a single composite score. Although this facilitates benchmarking, it may oversimplify the inherently multidimensional nature of DT. To address this, our proposal foresees the adaptation to different municipalities, allowing dynamic weighting mechanisms and modular scoring systems that can be adapted to the strategic priorities of each municipality. That is to say, instead of the 70/30 distribution, different percentages could be parametrized or different modules could be formed with different dimensions when the organization considers it necessary for its aims in the digital transformation process.

Despite these areas for improvement, the integration of AI techniques and the combined use of structured and unstructured data represent a methodological innovation with strong potential for scalability, automation, and replicability. The model’s validation through a real-world case study reinforces its practical value. The methodology followed in this paper in order to reach these contributions is detailed in the next section.

## 3. Materials and Methods

To address the research question of how to effectively and objectively assess the level of DT in local public administrations, this study adopts a methodology that integrates structured and unstructured data sources, combines AI-based analysis with interpretability tools, and supports decision-making. The methodology adopted in this study (Figure 1) is structured into four phases: (1) identification and integration of data sources; (2) preprocessing and transformation using extract–transform–load (ETL) procedures; (3) application of AI methods; and (4) development of decision-support dashboards and automated estimation of the Digital Transformation Index (DTI). This methodological structure is grounded in and extends the approach proposed in [37], which combines data integration, analytical processing, and visualization to support digital decision-making. Furthermore, this four-phase pipeline adheres to established data warehousing practices, aligning with ETL and analytical workflows in data warehousing [38]. In addition, our ingestion layer implements recent patterns for automated Internet of Things (IoT) data onboarding and schema harmonization, thereby enabling reproducible visualization and analytics [39].

Phase 1—Data sources and corpus construction. The primary data source consists of a series of reports developed by the Digital Intelligence Center of Alicante (CENID) based on survey data, which provide a comprehensive overview of the DT process in local public administrations within the province of Alicante (Alicante is one of the three provinces that make up the Valencian Community (Comunidad Valenciana), an autonomous region in southeastern Spain. The other two provinces are Valencia and Castellón), Spain. In particular, the ASIS (Análisis y observación del estado digital de los municipios de la provincia de Alicante, Analysis and observation of the digital status of municipalities in the province of Alicante) report analyzes the degree of progress in DT across 79 municipalities [40]. Several key areas and indicators related to digital transformation are examined, including communication infrastructures, human resources, Smart City initiatives, and smart tourist destinations, among others. Specifically, within the latter two categories, particular attention is given to vertical domains that, due to their relevance, incorporate IoT technologies as essential components for their deployment and operation. For each of the organizations included in the report, a Digital Transformation Index (DTI) was calculated using a novel index developed in the context of this study (previously introduced). The index is described in detail in Section 4.1 In addition to the structured data collected through surveys, a complementary corpus was created by extracting textual content from the official websites of the local public administrations in the province of Alicante that were included in the report. This unstructured data was linked to the corresponding DTI values, allowing both types of information to be used in the experimentation phase. State-of-the-art surveys consolidate IoT-centric Smart City stacks that couple edge/cloud dataflows with ML/AI, supporting our modular architecture [41].

Phase 2—Data engineering (ETL) and text preprocessing. A series of ETL processes was applied to integrate and clean data from multiple heterogeneous sources. Standard text analytics (preprocessing—tokenization, stop-word removal, and normalization) supports robust downstream modeling [42]. This step ensured the quality and consistency of the dataset for later analysis. Textual content extracted from municipal websites was preprocessed to remove non-relevant elements such as HTML (HyperText Markup Language) tags, URLs (Uniform Resource Locators), and identifying references. The cleaned content was then converted into a structured format and merged with the corresponding survey data. The choice of transformer encoders is aligned with current, journal-level syntheses of pre-trained language models and their fine-tuning regimes for downstream tasks [43].

Phase 3—AI-based estimation of the DTI. Two types of models were used: a neural network trained on the structured survey responses, and a transformer-based language model fine-tuned on the unstructured web content of each municipality. For structured (survey) signals, recent journal surveys on transformer representations for tabular data motivate our unified modeling strategy [44]. These models were evaluated to determine whether DT levels could be reliably predicted using distinct input types. The experimental setup and evaluation results are presented in Section 4.

Phase 4—Decision-support dashboards. To support decision-making and facilitate interpretation of the results, a series of dashboards was developed to visualize Key Performance Indicators (KPIs). These dashboards provide a graphical synthesis of the data, enable the continuous monitoring of digital maturity, and allow customization of indicators based on the characteristics of each municipality. As noted by the authors of [45], data visualization is a process for conveying the importance of data through visual context, and it forms part of data analytics, executed after data correction. Several Business Intelligence (BI) platforms, such as Tableau, Looker Studio, and Power BI, support the development of such dashboards. Power BI was selected for this project as the primary tool. In the context of local public administrations, BI platforms are essential for enhancing transparency, optimizing resource management, and supporting the design of data-driven public policies, while also facilitating citizen access to public information in an accessible and meaningful way. Embedding what-if analysis into NGSI-compliant dashboards enables policy exploration and KPI-based impact assessment across urban domains [46].

## 4. Experiments

This section reports the experiments designed to assess the effectiveness of AI methods for the automated estimation of the DTI in public-sector organizations. The experimental evaluation leverages two data modalities from local public administrations: structured survey responses and unstructured textual content harvested from the official websites of the same organizations, which are jointly exploited to train and evaluate the proposed models.

Although the empirical focus is municipalities in the province of Alicante, the methodological approach is portable to other geographical and institutional settings. Experimental studies conducted across heterogeneous socio-economic contexts have questioned the one-size-fits-all applicability of traditional Smart City models, underscoring the need to localize DT frameworks to local realities [47]. Complementary research within specific national public administrations has documented DT trajectories shaped by institutional and cultural factors, revealing both challenges and opportunities for implementation [48]. Moreover, deployments of urban smart technologies have prompted critical debates on the ethical dimensions of data use, inclusion, and governance—issues that must be addressed when transferring models across contexts [49]. Taken together, these perspectives support the adaptability, external validity, and careful contextualization of the proposed framework for municipalities operating under diverse structural conditions.

The first subsection provides a detailed description of the data used in this study, along with the definition of the newly proposed DTI. The second subsection describes the training of a neural network model based on the survey data, with the objective of determining whether the model can effectively learn to approximate the DTI of a local public administration. The third subsection presents the fine-tuning of a transformer-based model on the official websites of the organizations that completed the surveys in order to assess whether their web content contains sufficient information to estimate their DTI. The section concludes with three analytical subsections: the first examines the experimental results; the second presents the development of interactive dashboards; and the third outlines future directions for integrating sensor-based data into the proposed framework.

### 4.1. Data Description

This section provides a detailed description of the structured and unstructured data used in our experiments. In addition, it introduces the formula used to compute the DTI. The construction, normalization, and weighting of composite indicators follow internationally recognized guidance to ensure interpretability and robustness [50].

#### 4.1.1. Structured Data from Municipal Interview Surveys

The primary source of structured information used in this study is the ASIS report, which contains survey data collected by CENID from 79 municipalities in the province of Alicante during the years 2020 and 2021. This report analyzes the degree of progress in the digitalization of local public administrations in Alicante (Spain). The survey results are stored in spreadsheet files (Excel format), with each municipality completing one survey composed of 93 main questions and subquestions. These questions are classified into nine core components: Workstation; Communication Infrastructure; Human Resources; Backoffice ICT Infrastructure; Frontoffice ICT Infrastructure; Smart City; Smart Tourist Destination; Municipal Action Plans; and Citizen Perception. The survey responses vary in format and include closed-ended questions, open-ended questions, multiple-choice items, and combinations of these types.

The questionnaire design used in the ASIS report followed several key characteristics:The questions are mainly closed-ended, which facilitates statistical processing and ensures consistency in responses. In addition, rating scales (e.g., from 1 to 5, or “Not at all” to “Very much”) were used to measure perceptions and levels of implementation.The questionnaire covers a broad range of dimensions related to the digital transformation of municipal services, including innovation, governance, technologies used, citizen participation, and digital competencies of municipal staff. This provides a multidimensional perspective to the study.The questions were tailored to the size and competencies of each municipality to ensure relevance for both large and small/medium-sized municipalities. Stratification by population size was used to create representative segments of the study universe.The survey was addressed to technical staff and municipal managers, thereby collecting information from those directly involved in the implementation of digital services.The dataset includes both quantitative indicators (e.g., human and technological resources, use of electronic services, and process automation levels) and qualitative indicators (e.g., perceived barriers and digital maturity levels).To ensure a high participation rate and accurate responses, the survey was conducted using a face-to-face strategy. The protocol consisted of the following:–Sending a formal notice to the mayors of selected municipalities to inform them about the study and to request collaboration by identifying key informants—individuals with appropriate expertise and knowledge.–Holding an in-person meeting at the University of Alicante with selected participants to explain the study and methodology in detail.–Scheduling appointments with the designated respondents in each municipality for the administration of the questionnaire.

#### 4.1.2. Unstructured Data Collection and Corpus Creation

In addition to the structured survey data, a complementary dataset was constructed using unstructured content extracted from the official websites of the local public administrations included in the ASIS report. The aim was to build a textual corpus suitable for training and evaluating (AI) models capable of estimating each organization’s DTI.

The web content for each municipality was downloaded using recursive crawling tools that allowed for controlled retrieval of publicly available pages, specifically limiting the depth of navigation and restricting the file types to HTML content. Multimedia elements, scripts, and style sheets were intentionally excluded to reduce noise and focus on textual information relevant to the digital maturity of each entity.

Once downloaded, the HTML files were converted into plain text format to extract readable content. This transformation was carried out using text-based rendering tools that emulate a browser’s view of the page and discard non-informative tags and metadata.

A preprocessing phase was then performed to clean the textual data. This included the following:Removal of HTML tags, URLs, and special characters;Anonymization of municipal names and demonyms to prevent model bias;Normalization of accents, casing, and spacing inconsistencies.

The cleaned content of each municipality’s website was then consolidated into a single .json file. These files were subsequently merged with the corresponding survey data for each organization. The result was a unified dataset in which each instance included both the structured responses and the associated unstructured web content. The unified dataset and code used in this study are openly accessible to the research and technical community via GitHub: https://github.com/maa104/Digital-Transformation-LocalAdmin (visited on 4 July 2025).

This combined dataset was used as the input for the experiments described in the next sections, enabling the training and evaluation of both neural network and transformer-based models for automated DTI prediction.

#### 4.1.3. The New DTI

As previously mentioned, this work proposes the development of a new DTI: a weighted, multidimensional formula specifically designed to assess digital maturity in local governments. The index draws inspiration from well-established models such as DESI and the Corporate DTI [23], while incorporating dimensions unique to the municipal domain—such as Smart City platforms and smart tourism services—which are typically absent from generic models but essential in local contexts [29].

The design of the DTI follows three core principles:Holistic approach: It integrates technical (infrastructure), operational (services), and strategic (Smart City) dimensions, in alignment with the literature emphasizing the multifactorial nature of DT [51];Differentiated weighting: It assigns variable weights to each dimension based on its relative impact, prioritizing those with greater influence on citizen engagement and service delivery;Contextual adaptability: It includes indicators tailored to the local public sector, allowing for more precise and context-sensitive assessments.

The DTI is structured into two weighted layers, combining general digitalization metrics with context-specific indicators (Figure 2). This design provides both flexibility and methodological rigor for assessing DT in municipalities. The proposed model includes a general core (70%) and a contextual specialization layer (30%). However, these percentages are not fixed. The methodology is designed to parametrize and adjust these weights, allowing the model to be adapted to the strategic priorities and characteristics of each organization—whether a local public administration, healthcare provider, educational institution, or industrial company. In each case, both the general and specific components can be redefined to reflect the requirements of the domain under study. This adaptability makes the DTI a versatile and scalable tool for evaluating digital maturity across diverse institutional environments.

##### Generalizable Core Dimensions (70%)

This foundational component captures DT elements common across organizations and sectors. These dimensions are essential for establishing a baseline of digital readiness and operational capacity, and they facilitate inter-organizational benchmarking:Communication Infrastructure (10%): Assesses the quality of network infrastructure and connectivity, a foundational prerequisite for DT;Backoffice (10%): Measures the digitalization of internal administrative processes, essential for operational efficiency;ICT Equipment (20%): Evaluates the modernization of technological assets, reflecting the institution’s digital capacity;Digital Services (20%): Quantifies the availability and accessibility of online public services, a key indicator of digital maturity [29];Strategic Planning (10%): Assesses the existence, regular updating, and digitalization of strategic and operational plans (e.g., emergency response, sustainability, mobility, or digital inclusion), reflecting proactive and resilient governance.

This framework, composed of five core dimensions, establishes a versatile digital backbone. This structure is designed to be a robust and comparable tool for evaluating DT across a wide range of public and private organizations. By providing a consistent and methodologically sound approach, it ensures that assessments are both reliable and comparable, regardless of the organization’s sector or size.

##### Context-Specific Dimensions for Local Public Administration (30%)

In addition to the core framework, the DTI incorporates two specialized dimensions to accurately capture the unique characteristics and strategic priorities of local governments. These dimensions are crucial for reflecting the vital role municipalities play in both smart governance and territorial innovation.

Smart Cities (20%): Includes the deployment of IoT platforms and vertical solutions (e.g., mobility or energy), which are foundational to intelligent urban ecosystems;Smart Tourism Destination (10%): An innovative dimension that evaluates digital services aimed at visitors, particularly relevant in tourism-driven municipalities.

These context-specific dimensions ensure that the DTI not only captures general digital maturity but also aligns with the territorial and functional mandates of local administrations, offering a more nuanced and policy-relevant assessment.

Although the DTI was originally designed for local public administrations, its modular, two-layer architecture facilitates adaptation to other domains, including the industrial sector. The generalizable core dimensions remain applicable because they capture capabilities that underpin digital maturity across contexts. By contrast, the context-specific dimensions should be reframed to reflect domain priorities. For instance, in industrial settings, the Smart City and Smart Tourism dimensions could be replaced by Industrial Automation Readiness (e.g., IoT-enabled production lines; robotics) and Digital Supply Chain Integration (e.g., ERP–MES integration to enable real-time logistics and production control).

This flexibility strengthens the overall robustness of the framework and supports cross-sector deployment. By separating stable, domain-agnostic dimensions from configurable, sector-tailored indicators, the two-layer design enhances scalability, interpretability, and replicability, enabling rigorous DT benchmarking across diverse organizational contexts while preserving sensitivity to local or sector-specific dynamics.

Such flexibility is consistent with current digital-transformation scholarship showing that modular, multi-layered designs generalize across sectors while preserving domain specificity [51,52,53]. In our case, the dual-structure (domain-level dimensions + operational indicators) follows best practice in composite-indicator construction—explicit about normalization, weighting, and aggregation—thus supporting robustness and replicability under alternative model choices [54,55]. Evidence from maturity-modeling and benchmarking studies further indicates that layered indices scale from firm- and sector-level use cases to cross-industry comparisons without losing interpretability [56,57]. Finally, recent advances on spatial and uncertainty-aware composite indicators show how local structure, dependence, and estimation error can be incorporated transparently, which strengthens sensitivity to local or sector-specific dynamics while maintaining comparability [58,59].

### 4.2. Neural Network for DTI Prediction from Survey Data

This subsection details the neural network (NN) model trained on the numeric fields extracted from the survey responses. Specifically, the model uses 67 fields related to digital infrastructure, services, and Smart City indicators, including the following: “Fiber Optic, Copper Links, Inter-site Radio Link, Internet Speed, Adequate Speed, Internet Redundancy, Proprietary Infrastructure, Available 4G Coverage, Available 5G Coverage, Municipal Fiber Availability, Interconnection, Data Processing Center (DPC), DPC Utilization, Firewall, Antivirus Software, Antispam System, Denial of Service System, Attacks, Theft, Delegate Authority, Office Management, Databases, Employee Portal, Document Management, Accounting Management, Electronic Signature, Human Resources Management, Control and Monitoring, Municipal Asset Management, Population Register Management, Grant Management, Geographic Information System (GIS), Library, Emergency Services, Local Police, Traffic Management, Vehicles, Transportation, Street Vendors, Markets, Funeral Activities, Sports, Culture, Education, Gender-Based Violence, Buildings, Construction Licenses, Road Incidents, Social Services, Average Years, Renewals, Electronic Portal, Website, Administrative Resources, Smart City Plan, Smart City Platform, Smart City Commission, Smart City Funding, RECI Membership, Tourism Plan, Smart Tourism Platform, Smart Tourist Destination (STD) Components, STD Alliances, Municipal Plans, Experience, Citizen Digitalization, and Citizen Digital Literacy”.

All fields are represented as integers with values ranging from 0 to 4. An excerpt from the dataset showing the input values and the corresponding target output (i.e., the Digital Transformation Index, DTI) is shown in Table 2.

#### 4.2.1. Validation Method

To assess the predictive performance of each machine learning model, a resampling technique was employed—specifically, *k*-fold cross-validation. We adopted k-fold cross-validation consistent with recent methodological reviews emphasizing rigorous, bias-aware performance estimation [60]. This method involves partitioning the dataset into *k* disjoint subsets of approximately equal size. In each iteration, one subset is designated as the test set (25%, i.e., 20 surveys), while the remaining *k* − 1 subsets are used as the training set (75%, i.e., 59 surveys). This process is repeated *k* times, resulting in *k* distinct models. The final performance metric is computed as the average of the values obtained across all iterations. In this study, the value of *k* was set to 10.

#### 4.2.2. Neural Network Architecture

The optimal configuration was achieved with a four-layer feedforward network. The architecture is summarized as follows:Input layer: 128 neurons;Hidden layers: 64 neurons and 32 neurons, both with ReLU activation;Output layer: 1 neuron with linear activation (predicting a float value between 0 and 100);Loss function: Mean Squared Error (MSE), suitable for regression tasks.

The layer configuration and total parameters are shown in Table 3.

#### 4.2.3. Results

We report MAE (with RMSE) in line with current guidance on metric selection: MAE for Laplace-like error profiles and RMSE for Gaussian regimes [61]. After 1000 training epochs, the neural network achieved the following results:Mean Absolute Error (MAE) on training set: 0.050;MAE on test set: 7.788.

These results demonstrate that an NN can successfully approximate the DTI from survey data and provide a reliable tool for automating the evaluation of DT in local public administrations.

### 4.3. Transformer Model Trained on Organizational Web Content for DTI Prediction

This subsection explores an alternative approach to using survey data, where the web content of the organizations is leveraged to estimate their DTI. To this end, a corpus was constructed using the official websites of the organizations that completed the surveys, paired with their previously calculated DTI values. This dataset was used to fine-tune a transformer model that predicts a DTI score.

The model selected for this task is bertin-roberta-base-spanish (https://huggingface.co/PlanTL-GOB-ES (visited on 4 July 2025)), which is a transformer-based masked language model for the Spanish language (since Spanish is the language of the text to be processed, although any BERT-based model would be fit for the task if it is properly adjusted for the regression task and the language of the texts). It is based on the RoBERTa large model and has been pre-trained using the largest Spanish corpus known to date. The model was fine-tuned to output a float value between 0 and 100. An excerpt of the Python code (version 3.13.1, available online: https://www.python.org (accessed on 4 July 2025)) used for the fine-tuning is presented next:


    training_args = TrainingArguments(



        output_dir="./results",



        num_train_epochs=NUM_TRAIN_EPOCHS,



        per_device_train_batch_size=BATCH_SIZE,



        per_device_eval_batch_size=BATCH_SIZE,



        warmup_steps=500,



        weight_decay=0.01,



        logging_dir="./logs",



        logging_steps=100,



        evaluation_strategy="epoch",



        save_steps=500,



        save_strategy="steps",



        load_best_model_at_end=False,



        metric_for_best_model="mse",



        greater_is_better=False,



        learning_rate=LEARNING_RATE,



        report_to="none"



    )



    trainer = Trainer(



        model=model,



        args=training_args,



        train_dataset=tokenized_train_dataset,



        eval_dataset=tokenized_test_dataset,



        compute_metrics=compute_metrics,



    )



    trainer.train()



    eval_results = trainer.evaluate()


Table 4 shows a representative excerpt of the input-output pairs used during training. The first column contains the first 400 characters extracted from the HTML content of the municipalities’ websites, while the second column displays the corresponding DTI value.

#### Model Evaluation

To assess the performance of the fine-tuned transformer, 10-fold cross-validation was applied using the default mean MSE loss function. After 70 training epochs, the model achieved the following results:MAE on the training set: 0.424;MAE on the test set: 7.889.

These results indicate that a simple web content extraction approach can also provide a reasonable approximation of an organization’s DT level.

### 4.4. Analysis of Experimental Results

The experimental results obtained confirm the effectiveness of AI models—both NN and transformer-based architectures—in estimating the proposed DTI for local public administrations. Despite the limited size of the training corpus, the models achieved promising results, demonstrating their generalization capability. These findings support the validity of the proposed methodology and its applicability to other contexts.

In the case of NN trained on structured survey data, the model demonstrated a high level of predictive accuracy. The low MAE observed in both training and testing phases indicates that the model successfully captured the underlying patterns and relationships between digital infrastructure variables and the DTI. This validates the feasibility of automating the evaluation process based on structured inputs collected directly from organizations.

Similarly, the transformer model fine-tuned on the web content of municipalities also yielded highly promising results. Despite the unstructured and heterogeneous nature of the input data (i.e., raw HTML content), the model was able to approximate the DTI values with a high degree of reliability, thus enabling an automated estimation process that eliminates the need for survey-based data collection. This suggests that valuable information about the digital maturity of an organization can be inferred from publicly available web content, which opens the door to scalable, low-cost evaluation methods based on digital footprints.

These findings demonstrate that both AI approaches—applied to structured and unstructured data—are capable of generating accurate and interpretable predictions. The success of these models reinforces the validity of the proposed methodology and highlights the potential of combining traditional techniques with AI-driven approaches to assess DT in public sector entities. Consistent with recent evidence, strong learners (deep neural networks/transformers) tend to outperform standard benchmarks on text–tabular and sensor forecasting tasks, and multimodal fusion (structured and unstructured data) yields systematic gains over single-modality baselines. This behavior has been observed in domains such as healthcare, urban analytics, and sensor-based forecasting in Smart Cities [62,63,64,65,66,67,68,69]. These findings support the validity of our approach and reinforce the statistical robustness of the results obtained in the study.

### 4.5. Visual Analysis Through Dashboards

In the context of territorial DT, data visualization has become an essential tool for interpreting complex information and supporting strategic decision-making. Power BI enables the creation of interactive dashboards that consolidate multiple data sources and present KPIs in an accessible and dynamic format. This capability is particularly valuable for diagnostic studies like the ASIS project, which assesses the level of digital maturity across municipalities.

In this study, a set of Power BI dashboards was developed to visually represent the DTI scores of the 79 municipalities in the province of Alicante included in the sample. Each dashboard is organized around a specific analytical dimension—such as Communication Infrastructure or Backoffice—and incorporates tailored KPIs to facilitate accurate and contextualized analysis.

These dashboards provide a comprehensive view of each municipality’s digital maturity, support inter-municipal comparisons, and highlight areas in need of strategic intervention. By leveraging empirically grounded data, this approach establishes a robust basis for public policy formulation and the development of local digital strategies.

Among the most noteworthy dashboards is the one depicting the overall DTI through a geospatial visualization based on the proposed weighted formula (Figure 3). The results highlight that cities such as Alicante, Elche, Alcoy, Villena, and Benidorm demonstrate the highest levels of digitalization. This trend can be attributed to several structural and contextual factors commonly associated with larger municipalities, including greater financial and human resources, the presence of dedicated digital strategy units, and higher citizen demand for advanced digital services. In contrast, smaller municipalities often face limitations in infrastructure, technical expertise, and strategic planning capacity, which contribute to more pronounced deficits in their digital maturity.

The dashboard corresponding to the Smart City dimension (Figure 4) reveals that most municipalities have yet to implement strategic master plans (79.7% of the municipalities, 63/79) or dedicated digital platforms (89.9%, 71/79). However, there are scattered initiatives in certain locations—such as energy efficiency systems (78.5%, 62/79), digital tools for citizen participation (53.2%, 42/79), and smart irrigation in green areas (21.5%, 17/79). After analyzing the results, this dimension emerges as a promising yet still underdeveloped area within the broader DT process.

With respect to the Smart Tourist Destination (STD) dimension (Figure 5), notable advances include the availability of online tourism indicators (27.8% of the municipalities, 22/79), public Wi-Fi connectivity (25.3%, 20/79), and digital guided tours (20.3%, 16/79). However, the widespread absence of formal strategic planning (75.9%, 60/79) and specialized digital platforms (81.0%, 64/79) indicates that the development of STD initiatives remains in an early and uneven stage.

### 4.6. Challenges and Future Directions in Sensor Integration for Local Governments

The analysis of the experimental results reveals a marked deficiency in the deployment of sensor-based technologies within local public administrations of the Valencian Community, particularly within the Smart City domain. Despite the initiation of DT processes by many municipalities, the systematic integration of sensor networks remains notably limited. Strategic verticals such as smart parking, intelligent pedestrian crossings, and smart irrigation systems (Figure 4)—all reliant on sensor infrastructures—are still insufficiently implemented. Specifically, 79.7% of municipalities (63/79) lack dedicated Smart City strategic plans, while 89.9% (71/79) do not possess digital platforms designed for intelligent management. The adoption of sensor-enabled initiatives is equally sparse; only 21.5% of municipalities (17/79) have deployed smart irrigation systems in green areas, and although 53.2% (42/79) use digital tools to foster citizen participation, many of these tools are not linked to real-time sensor data. This low level of IoT integration impairs local governments’ ability to acquire real-time data, which is crucial for evidence-based decision-making. Consequently, operational efficiency and responsiveness to citizen needs are constrained, limiting the potential of these administrations to evolve into fully resilient and intelligent urban ecosystems. Addressing these gaps will be essential for advancing sensor integration as a cornerstone of sustainable and proactive Smart City development in the region.

Moreover, the findings emphasize the indispensable role of high-quality data in supporting robust ETL workflows, which are critical for the generation of decision-support dashboards that underpin data-driven public management. In this context, IoT-enabled sensorization constitutes a foundational component across the various verticals of a Smart City, facilitating the continuous collection and analysis of urban data.

The ASIS [40] report highlights that this technological gap is particularly acute in smaller municipalities, which often face financial and technical constraints. In these localities, the lack of investment in digital infrastructure and the shortage of qualified personnel impede the adoption of data-driven solutions, perpetuating inefficiencies and reducing administrative adaptability. These findings are consistent with the recent literature emphasizing the transformative potential of IoT and machine learning in fostering data-centric governance models aimed at improving public service delivery and urban quality of life [70].

To guide future development, a strategic framework is proposed to identify and classify the main categories of sensors that would be highly beneficial in accelerating the digital and smart transformation of municipalities. This classification aims to align sensor deployment with key urban functions and public service areas (Table 5). To operationalize sensorization in local governments, five critical application domains are identified, each corresponding to a Smart City vertical with significant transformative potential:
(1)Traffic and Mobility Control.

Efficient traffic and mobility management represents a core pillar in the development of Smart Cities, relying heavily on the deployment of sensor-based technologies and IoT systems. Within this Smart City domain, the integration of sensing technologies enables real-time monitoring and optimization of urban mobility through applications such as vehicle detection, traffic flow analysis, and dynamic tracking of public transportation systems. Typical implementations include the following:Vehicle counting sensors and automatic number plate recognition (ANPR) systems;Traffic and surveillance cameras for intersections and accident-prone zones;Occupancy sensors in public parking areas;Smart pedestrian crossings and real-time public transport monitoring.

The adoption of these systems can significantly contribute to congestion mitigation, enhanced road safety, and the advancement of sustainable mobility strategies. A diverse range of sensing technologies—such as inductive loops, CCTV cameras, acoustic sensors, and Bluetooth-based detection—are employed to collect high-resolution data on vehicular and pedestrian movements. These data streams are processed and integrated into intelligent traffic control infrastructures that support adaptive traffic light regulation, congestion management, and accident prevention mechanisms.

Ultimately, the integration of such sensor-based systems not only increases the operational efficiency of urban mobility networks but also facilitates the shift towards citizen-centric, sustainable mobility policies aligned with the broader goals of smart urban governance [19].

(2)Risk and Safety Monitoring.

Risk prevention and emergency response are critical components of resilience strategies in Smart Cities and can be substantially enhanced through the integration of sensor-based early warning systems. These technologies enable real-time detection and situational awareness by leveraging distributed sensing infrastructures across urban environments. Representative examples include the following:Motion and infrared sensors for crowd monitoring in public spaces;Heat and flame detectors in wildfire-prone zones;Water-level sensors and weather stations for flood detection;Air quality sensors in industrial or high-risk zones.

The deployment of such technologies significantly enhances a city’s ability to anticipate, detect, and respond effectively to critical incidents. Smart urban environments increasingly rely on real-time, geo-referenced data collected via IoT devices and sensor networks to provide high-value services in public safety and disaster risk management [71].

However, the implementation of these systems faces considerable challenges, including fragmented infrastructure ownership, limited access to sustainable funding, and persistent public concerns regarding privacy and surveillance. Addressing these barriers through inclusive governance frameworks and robust data management strategies is essential for empowering local governments to act promptly and efficiently during emergencies. The systematic integration of these sensing capabilities can serve as a foundational element in building adaptive, responsive, and resilient urban systems [71].

(3)Environmental Health Surveillance.

Environmental monitoring plays a pivotal role in advancing public health and promoting urban sustainability within the framework of Smart Cities. The deployment of advanced environmental sensors facilitates continuous, real-time data collection that supports evidence-based urban planning and proactive health risk management. Key sensor technologies include the following:Air pollution detectors (e.g., NO_2_, PM2.5, CO);Acoustic sensors to measure and map noise pollution;UV radiation and pollen level monitors for vulnerable populations;Weather and humidity sensors to inform energy and water consumption policies.

The integration of these technologies enables urban systems to detect and respond to environmental threats more efficiently, particularly when linked to urban health platforms. The convergence of IoT, AI, and 5G connectivity in smart medical and environmental devices has demonstrated significant potential not only in monitoring clinical parameters but also in enhancing environmental surveillance. These systems facilitate early detection of conditions affecting population health, especially among at-risk groups such as the elderly and individuals with chronic illnesses [71].

By enabling timely responses to adverse environmental events, such integrated platforms help optimize resource use and strengthen the resilience of urban communities. However, their implementation faces critical challenges, including disparities in access to digital infrastructure, limited device interoperability, and concerns over data privacy and protection. These barriers underscore the urgent need for robust regulatory frameworks and context-sensitive deployment strategies that ensure equitable access and sustainable operation of environmental monitoring technologies in smart urban ecosystems.

(4)Energy Monitoring and Management.

Energy monitoring and management systems are fundamental components in advancing the sustainability and operational efficiency of urban infrastructure, particularly within the framework of Smart Cities across the European Union. As European municipalities progress toward the objectives outlined in the European Green Deal and the Fit for 55 legislative (https://eur-lex.europa.eu/legal-content/ES/TXT/?uri=CELEX:52021DC0550 (visited on 4 July 2025)) package—which aims to reduce net greenhouse gas emissions by 55% by 2030 compared to 1990 levels (European Commission, 2021)—the deployment of sensor-based energy technologies becomes increasingly critical. These technologies enable a wide range of high-impact applications, including the following:Smart lighting systems that adjust their intensity based on ambient conditions or occupancy;Energy meters to track real-time consumption in public buildings;Solar panel and battery performance monitors;Load balancing sensors for optimizing electricity distribution and reducing peak demand.

The integration of IoT-enabled solutions in this domain fosters data-driven decision-making aimed at optimizing resource use and supporting the climate neutrality targets of the EU. The convergence of IoT infrastructure, cloud-based platforms, and advanced analytics not only facilitates the monitoring of energy consumption patterns but also enables the automation of decision-making processes related to energy efficiency and system optimization [72]. Their findings demonstrate that such integrated systems can significantly reduce operational costs, enhance asset longevity, and contribute to long-term environmental goals in urban contexts.

Lawande [73] highlights that the implementation of intelligent energy systems in urban environments supports predictive maintenance, energy demand forecasting, and dynamic resource allocation, thereby enhancing the resilience and adaptability of municipal services.

Nevertheless, the deployment of these systems within the European context faces several challenges, including fragmented multilevel governance, technical issues related to interoperability, and the complexity of integrating legacy infrastructure with emerging digital standards. Moreover, compliance with the General Data Protection Regulation (GDPR) (https://eur-lex.europa.eu/legal-content/EN/TXT/PDF/?uri=CELEX:32016R0679 (visited on 6 July 2025)) introduces a critical dimension concerning the ethical and secure management of energy-related data [74].

Addressing these barriers requires coordinated action through standardized protocols promoted by European standardization bodies such as CEN-CENELEC (https://www.cencenelec.eu (visited on 4 July 2025)), the promotion of public–private partnerships as mechanisms to scale strategic energy digitalization projects, and the strategic use of EU funding instruments, including Horizon Europe and the Digital Europe Program, which provide financial and technical support for initiatives that integrate smart energy and DT in urban settings (https://research-and-innovation.ec.europa.eu (visited on 4 July 2025)).

Advancing toward intelligent energy systems represents a pivotal step for European cities to become not only smarter but also more sustainable, inclusive, and resilient.

(5)Urban Cleanliness and Waste Management.

Maintaining a clean, safe, and sustainable urban environment increasingly relies on the deployment of intelligent waste monitoring and management technologies. These systems, which form a core component of Smart City infrastructure, enable real-time data collection, predictive analytics, and automated decision-making to optimize municipal waste operations. Key applications include the following:Fill-level sensors embedded in waste containers, which allow dynamic route optimization for collection vehicles, reducing fuel consumption and operational costs [70];Smart recycling stations equipped with usage counters, contamination detectors, and user feedback interfaces to enhance recycling efficiency and citizen participation [73];Environmental sensors capable of detecting illegal dumping, waste overflow, or hazardous emissions (e.g., methane or ammonia) in public areas, thereby enabling rapid response and enforcement [72];Temperature and gas sensors in waste storage facilities to monitor fire risk and ensure compliance with safety regulations.

These technologies not only improve the operational efficiency of municipal cleaning services but also contribute to broader sustainability goals by reducing greenhouse gas emissions, minimizing landfill dependency, and promoting circular economy practices. As highlighted by Ullah [70], the integration of IoT devices with machine learning algorithms enables predictive maintenance, anomaly detection, and adaptive resource allocation in waste management systems. Moreover, Liu et al. [72] emphasize that government involvement—through fiscal incentives and regulatory frameworks—plays a critical role in scaling up these innovations and ensuring their alignment with environmental policy objectives. However, challenges persist, including data interoperability, privacy concerns, and the need for standardized protocols across heterogeneous urban infrastructures. Addressing these barriers through cross-sectoral collaboration and robust governance mechanisms is essential to fully realize the transformative potential of smart waste management in sustainable urban development.

#### Analysis of Sensor Integration for Local Governments

The implementation of a structured sensor classification framework in local public administrations presents a dual challenge. On one hand, municipalities often face technological, financial, and organizational constraints, including limited budgets, fragmented legacy systems, and a shortage of technical expertise. On the other hand, the establishment of standardized data models, the assurance of sensor interoperability, and the resolution of privacy-related concerns demand coordinated governance strategies and regulatory alignment.

Despite these barriers, the integration of sensor-based technologies holds significant transformative potential. When strategically deployed, these systems can substantially enhance service delivery, environmental sustainability, and citizen satisfaction. Rather than being viewed as a cost, sensor deployment should be considered a long-term investment that positions municipalities at the forefront of smart governance and digital innovation.

The findings of this study underscore that sensorization remains an underdeveloped frontier in the broader context of DT. While many local administrations have initiated digital initiatives, the systematic adoption of sensor technologies—particularly in small and resource-constrained municipalities—remains limited, as evidenced by the ASIS study. Addressing this gap through a clear, structured, and prioritized sensor deployment roadmap could catalyze the transition from digitally enabled municipalities to fully functional Smart Cities.

The proposed sensor classification framework, which encompasses key Smart City verticals such as mobility, public safety, environmental health, energy management, and waste monitoring, offers a strategic tool for aligning technological deployment with local operational and social priorities. This alignment is essential for maximizing the return on investment in digital infrastructure and for achieving broader goals related to sustainability, efficiency, and public value.

Empirical evidence supports the effectiveness of these technologies. For instance, ref. [70] demonstrates how the integration of IoT sensors with machine learning algorithms can optimize waste management through predictive maintenance and dynamic resource allocation. Similarly, ref. [72] emphasizes the enabling role of government involvement via fiscal incentives and regulatory frameworks in scaling sustainable technological solutions. In the energy domain, ref. [73] illustrates how intelligent monitoring systems can enhance urban resilience by enabling demand forecasting and automated maintenance processes.

However, the deployment of these systems is not without complexity. Key challenges include infrastructure fragmentation, lack of interoperability across devices, limited technical capacity, and public concerns regarding surveillance and data protection. In the European context, compliance with the General Data Protection Regulation (GDPR) adds an additional layer of complexity to the ethical management of sensor-generated data [74].

Overcoming these challenges requires a multi-stakeholder approach involving public authorities, private sector actors, and civil society. Standardization efforts led by organizations such as CEN-CENELEC (2023), the promotion of public–private partnerships (European PPP Expertise Centre, 2020, https://www.eib.org/en/products/advisory-services/epec/index (visited on 4 July 2025)), and the strategic use of EU funding instruments such as Horizon Europe and the Digital Europe Programme (https://research-and-innovation.ec.europa.eu/funding/funding-opportunities/funding-programmes-and-open-calls/horizon-europe_en (visited on 4 July 2025)) are critical to ensuring the scalability and long-term viability of smart sensor initiatives.

In conclusion, sensorization should not be regarded as an end in itself, but rather as a strategic enabler of more efficient, inclusive, and evidence-based public policies. Its strategic deployment has the potential to reshape how municipalities manage resources, engage with citizens, and respond to the complex challenges of the 21st century. It is therefore recommended that local governments adopt a context-sensitive, phased roadmap for sensor integration that supports their progression toward fully realized Smart City models.

## 5. Conclusions

This study proposes an innovative and replicable methodology to automatically assess the digital transformation (DT) level of local public administrations. Unlike traditional assessment approaches, which often rely solely on surveys or expert judgments, our method combines structured survey data with unstructured web content, and it leverages Artificial Intelligence (AI) models—including neural networks and transformer architectures—to estimate the Digital Transformation Index (DTI).

The results obtained from the real-world case study conducted in the province of Alicante (Spain) demonstrate the feasibility and effectiveness of this approach. The AI models trained on survey data and municipal websites yielded promising results—even with a limited training corpus—showing that digital maturity can be estimated using both structured and unstructured information. In addition, the creation of a domain-specific corpus—made openly accessible to the research community—and the development of interactive dashboards contributed to a more interpretable and actionable evaluation process. These tools not only help identify critical weaknesses in digital infrastructures and services but also support data-driven decision-making and continuous improvement in public sector organizations.

The methodology also emphasizes dimensions that are often overlooked in generic digital maturity models—such as Smart City platforms and smart tourism services—making it particularly suited to the local government context. Furthermore, the proposed approach is adaptable to other domains, such as the industrial or healthcare sectors.

### 5.1. Practical Implications

The proposed methodology has several practical implications for public sector organizations aiming to accelerate their DT. First, it provides a scalable and low-cost diagnostic tool that can be implemented without the need for exhaustive technical audits or large-scale survey campaigns. By combining AI-based analysis of web content with structured survey data, public administrations can estimate their level of digital maturity in an automated, replicable, and interpretable manner.

Second, the development of interactive dashboards enables decision-makers to visualize performance indicators, identify gaps, and prioritize strategic actions. These dashboards can be customized to reflect the operational realities of each municipality, supporting targeted interventions and performance monitoring over time.

Third, the new framework for sensor integration offers a clear roadmap for advancing Smart City initiatives. By aligning technological deployment with key urban service areas—such as mobility, safety, environment, energy, and cleaning—the framework helps local governments make informed investments that enhance public service delivery, sustainability, and citizen engagement.

Finally, the methodology is flexible enough to be adapted to other domains, including healthcare, education, or industry, thereby broadening its utility and facilitating cross-sectoral benchmarking.

### 5.2. Limitations and Future Research

Despite the promising results, this study has certain limitations. The experimental validation was conducted within a specific regional context, focusing on a limited number of municipalities in the province of Alicante (Spain). Consequently, further research is needed to assess the generalizability of the findings across broader geographic areas and diverse organizational settings.

Future work will also consider the integration of additional data sources—such as open data portals, social media content, or sensor-generated information—and the refinement of AI models to improve the interpretability and robustness of DTI predictions. Moreover, expanding the model’s scope to explicitly incorporate sustainability and governance frameworks—particularly Environmental, Social, and Governance (ESG) criteria and the United Nations Sustainable Development Goals (SDGs)—would enhance its capacity to guide more inclusive and responsible DT strategies.

Moreover, our proposal proposes a 70/30 distribution between general and context-specific components. However, it allows dynamic weighting mechanisms and modular scoring systems to adapt to the strategic priorities of each municipality. Even so, future versions of the model will explore the use of formal sensitivity analysis to enhance the robustness of the model and ensure its relevance across diverse municipal contexts.

In summary, this work contributes to the literature on DT assessment in the public sector by proposing a scalable, interpretable, and adaptable methodology that combines structured and unstructured data with advanced AI techniques. The resulting tools, including interactive dashboards and domain-specific indicators, support public administrations in designing effective, transparent, and citizen-centered digital strategies. Aligning future developments with international sustainability standards will further strengthen the model’s relevance for building equitable and resilient Smart Cities.

## Figures and Tables

**Figure 1 sensors-25-05179-f001:**
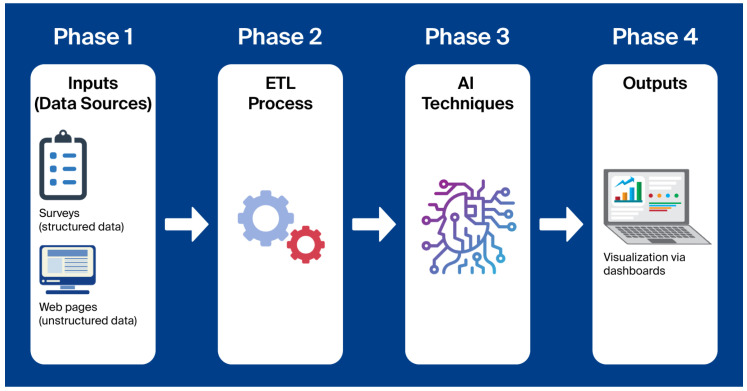
Proposed architecture for the automatic estimation of the DTI and the generation of decision-support dashboards.

**Figure 2 sensors-25-05179-f002:**
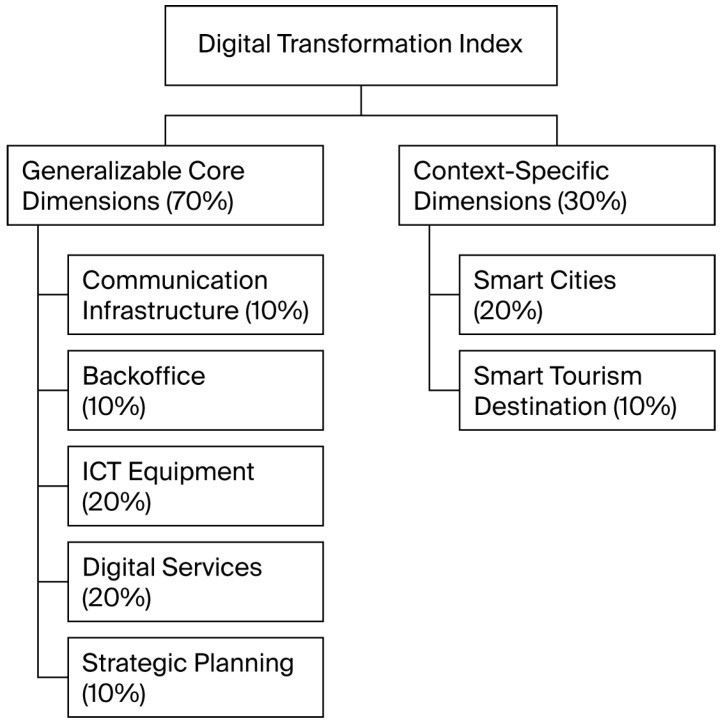
Two-layer structure of our proposed DTI, combining core and context-specific dimensions.

**Figure 3 sensors-25-05179-f003:**
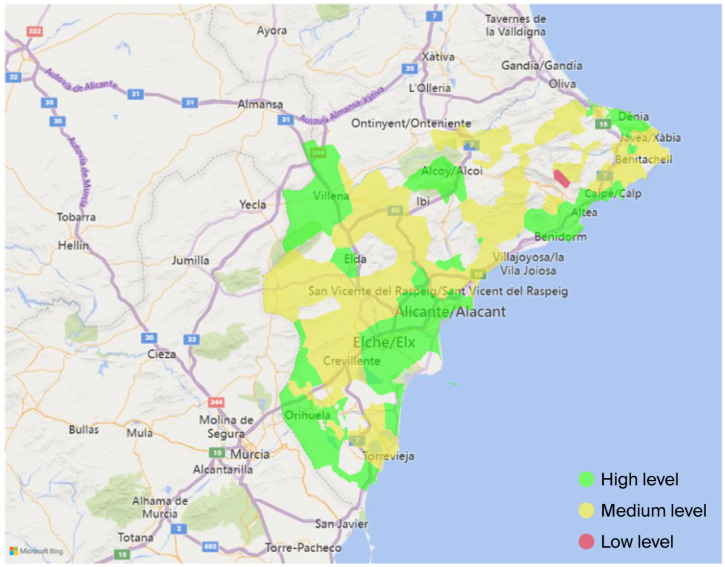
Summary of the DTI situation in the municipalities of Alicante during the years 2020 and 2021.

**Figure 4 sensors-25-05179-f004:**
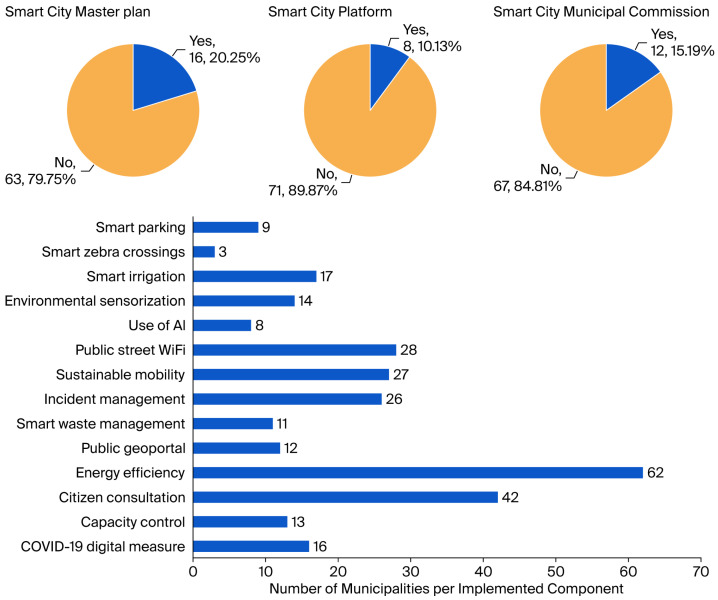
Smart City dimension dashboard.

**Figure 5 sensors-25-05179-f005:**
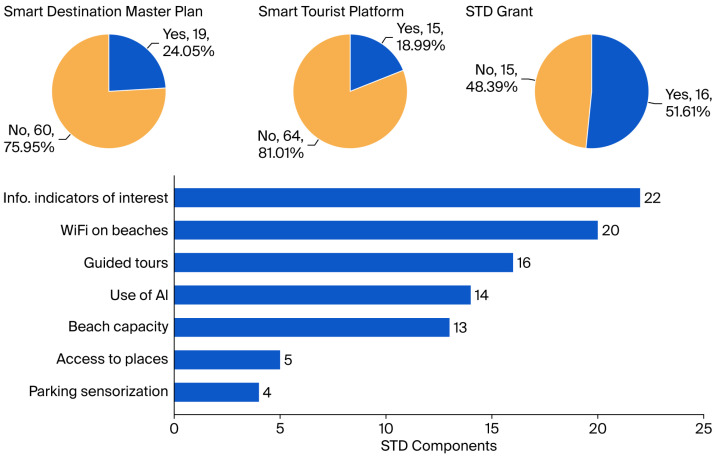
Smart Tourist Destination dimension dashboard.

**Table 1 sensors-25-05179-t001:** Summary of key digital transformation indices and maturity models.

Name	Type	Main Dimensions	Application Context
Digital Transformation Index (Metarius)	Index	Technological infrastructure, Business process digitalization, Leadership and culture, Digital customer experience, Data security and privacy	Private organizations; general benchmarking
Corporate Digital Transformation Index [23]	Index	Strategic leadership, Technological drive, Organizational empowerment, Environmental support, Digital outcomes, Applied digital technologies	Corporations and firms undergoing digital transformation
Conceptual Formula [24]	Conceptual Index	Technology, Customers, Human factor, Speed, Value, Need, Communications	Theoretical and exploratory models of DT
Digital Economy and Society Index (DESI)	Composite Index	Human capital, Connectivity, Integration of digital technologies, Digital public services	EU Member States; national and regional policy assessment
InAsPro Model [26]	Maturity Model	Technology, Organization, Social aspects, Strategy	Industrial sector; manufacturing firms
DX-MM Model [27]	Maturity Model	Strategic alignment, Organizational culture, Digital competencies, Business process maturity	General organizational readiness; self-assessment
BPM-based Model [28]	Maturity Model	Critical success factors, Process structure, Knowledge intensity, Strategic alignment	BPM contexts; knowledge-intensive and complex process environments
ODTR Model [29]	Maturity Model	Technological, Operational, Leadership, Human, Cultural	Organizational DT readiness; public and private sectors

**Table 2 sensors-25-05179-t002:** Excerpt of survey-based input values and corresponding DTI values.

Fiber Optic	Copper Links	Radio Link	Intern. Speed	Adeq. Speed	Intern. Redund.	Prop. Infr.	4G Coverage	DTI
1	2	2	1	2	2	2	1	50.89
1	2	2	1	2	2	2	1	50.89
1	2	1	2	1	2	3	1	49.50
1	2	1	2	1	2	3	1	57.91

**Table 3 sensors-25-05179-t003:** Neural network architecture (model: “sequential”).

Layer (Type)	Output Shape	Param #
dense (Dense)	(None, 128)	9472
dense_1 (Dense)	(None, 64)	8256
dense_2 (Dense)	(None, 32)	2080
dense_3 (Dense)	(None, 1)	33
Total params		**19,841**
Trainable params		**19,841**

**Table 4 sensors-25-05179-t004:** Examples of web content input and corresponding DTI values.

Web Text (Excerpt)	DTI
**Ajuntament de** ajuntament de de los comentarios ajuntament de ical@.es facebook × instagram rss facebook × instagram rss espanol valencia english ajuntament de inicio **noticias** saluda del **alcalde estructura municipal** o **junta de gobierno local o corporacion municipal o concejalias ordenanzas, tasas e impuestos registro de programas y aiu o agrupacion de interes** ur …	72.88
De de de los comentarios de ical de comentario **patrimonio cultural** del saltar al contenido espanol de menu de menu inicio **noticias** el la corporacion **plenos municipales planes, ordenanzas y reglamentos ayudas recibidas obras y urbanismo tesoreria juzgado de paz centro social biblioteca formacion servicios directorio telefonos transportes informes meteorolo** …	44.19
De los comentarios ical saltar al contenido espanol valenciano english menu menu inicio **noticias** el **saluda del alcalde corporacion municipal concejalias informacion al ciudadano o ordenanzas municipales o reglamentos o ofertas y bolsas de empleo o anuncios tramites y gestiones turismo** que visitar? **eventos rutas turisticas y actividades alojamiento gastronomia** pe …	46.32
**Ajuntament** de ajuntament de de los comentarios ajuntament de ical logo ajuntament de valencia espanol inicio saluda de **lalcalde agenda institucional corporacion municipal grupos municipales areas adl agencia de promocio del valencia biblioteca educacion esports juventud medio ambiente omic participacion ciudadana policia local servicios sociales turismo** urbani …	50.91

**Table 5 sensors-25-05179-t005:** Proposed classification of sensor categories for Smart City deployment in local governments.

Sensor Category	Examples and Applications
Traffic and Mobility Control	Vehicle counting, ANPR systems, traffic cameras, smart parking sensors, public transport tracking
Risk and Safety Monitoring	Motion detectors, crowd sensors, fire and flood sensors, weather stations, air quality alerts in risk zones
Environmental Health Surveillance	Air pollution sensors (NO_2_, CO, PM2.5), noise monitoring, UV/pollen level detectors, humidity/temperature sensors
Energy Monitoring and Management	Smart lighting, energy meters in public buildings, solar panel monitors, load balancing, and consumption tracking
Urban Cleanliness and Waste Management	Fill-level waste sensors, smart recycling points, illegal dumping detectors, gas/temperature sensors in waste containers

## Data Availability

The dataset and code used in this study are openly accessible via GitHub: https://github.com/maa104/Digital-Transformation-LocalAdmin (visited on 4 July 2025).

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
