# Peer review of "A Data-Driven Framework for Digital Transformation in Smart Cities: Integrating AI, Dashboards, and IoT Readiness"

_sensors, 2025, doi:10.3390/s25165179_

Round 1

Reviewer 1 Report

Comments and Suggestions for Authors

The work proposes an innovative methodology for automatically assessing the level of digital transformation (DT) in local public administrations, combining surveys and institutional website analysis using artificial intelligence (AI) techniques.
The paper is well-written, and the sequential presentation of the topics facilitates comprehension. The conceptual foundations are well-established, supported by the literature.

The approach of merging traditional methods and AI models to measure the level of DT is robust and original, standing out for proposing automation and scalability for diagnosing digital maturity in the public sector.

Suggestions:
Also highlight, in the abstract, more explicitly, the contextual limitations and the potential for international expansion of the model beyond Spain, to enhance the applicability of the proposed method.

Avoid excessive description of local policies or initiatives in the introduction, focusing on the innovative aspects of the work compared to international literature.

The study uses only municipalities in the province of Alicante as its experimental universe, which limits the generalizability of the results. Empirical validation is relevant, but further studies in different socioeconomic contexts are recommended.

The weighting of the DTI components (70% core/30% context) is based on expert consultation, but lacks formal validation and could benefit from systematic validation, such as sensitivity analysis.

Predictive models (NN, Transformer) present acceptable MAE, but the results should be explicitly compared with alternative benchmarks/treatments to reinforce statistical robustness.

Reviewer 2 Report

Comments and Suggestions for Authors

Please refer to my comments. The manuscript needs a review to improve scientific language, the clear highlighting of the study`s progress and added value.

Comments on the Quality of English Language

Please refer to my comments. The manuscript needs a review to improve scientific language, the clear highlighting of the study`s progress and added value. Please shorten text and improve the text flow for the readers. 

Reviewer 3 Report

Comments and Suggestions for Authors

This paper proposes a methodology combining traditional assessments with AI techniques (e.g., neural networks, transformers) to automatically evaluate digital transformation (DT) levels in public administrations. It uses staff surveys and AI-driven analysis of organizational data, validated via a case study in Spain’s Valencian Community. Results show that integrating IoT, sensor networks, and AI analytics can enhance sustainable Smart City development by improving efficiency and citizen-centric services.

Overall, I find this study interesting. However, during my reading, I encountered several points of confusion:

  • Model Justification: What is the rationale for using the tf_roberta_for_sequence_classification model?
  • Data Utilization: How are unstructured data utilized for neural network training? Could the authors clarify the basic structure of the data item in the corpus mentioned in the Introduction? Are examples of the data available?
  • Interpretation of Results: The statement in the paper—“The analysis of the experimental results reveals a significant shortcoming in the implementation of sensor technologies within local public administrations in the Valencian Community”—requires clarification. How does the paper specifically demonstrate this underutilization of sensor technologies by governments?
  • Context-Specific Training: The proposed DTI incorporates Context-Specific Dimensions. Does this imply that training must include context-specific data? If so, would data collection and retraining be prerequisites before evaluating a new, previously unseen dimension?

Reviewer 4 Report

Comments and Suggestions for Authors

Dear authors, here are my review's comments:

1) you say in your point 2.1 "..To address this, future versions of the model will explore the use of dynamic weighting.."

this should be put as Future research agenda (which normally is a point of the section Conclusions)

Also, clarify what do you mean by dynamic weighting and modular scoring..

In this same point add the rest: "..further empirical testing in broader geographic and institutional contexts will be essential to confirm its generalizability and to refine its applicability across a wider range of public sector environments".

Instead of these ideas, have a paragraph for introducing your methodology (which follows it in the next section of Materials and Methods)

2) So, Valencia belongs to Alicante?.. (clarify it please) in page 7 (where you mention CENID..)

what is 'ASIS' (report)? (didn't find..)

3) in Figure 2, why the two layers/dimensions have different weights?.. (70% and 30%) ? clarify it please 

4) in page 12, in the last paragraphs of point 4.1 (about the context-specific dimensions), use please recent references to support what you say there

 5) clarify please the fine-tuning process you refer (reflected in Table 4); was it done with python code?

6) section Conclusions should have subpoints such as Practical Implications; Limitations; and Future Research Agenda.

Round 2

Reviewer 4 Report

Comments and Suggestions for Authors

I checked that the authors have done their homework . Best regards, SF